# Influence of a hyperglycaemic diet on *Trypanosoma cruzi* infection in mice model

**Aline Coelho das Mercês¹, Flávia de Souza Marques¹, Bruno Teixeira Martins¹, Gabriel José Lucas Moreira², Bruno Mendes Roatt², Cláudia Martins Carneiro²,³, Silvia de Paula Gomes⁴, Joana Ferreira do Amaral⁵, Paula Melo de Abreu Vieira¹/⁺**

¹Universidade Federal de Ouro Preto, Instituto de Ciências Exatas e Biológicas, Núcleo de Pesquisas em Ciências Biológicas, Departamento de Ciências Biológicas, Laboratório de Morfopatologia, Ouro Preto, MG, Brasil
²Universidade Federal de Ouro Preto, Instituto de Ciências Exatas e Biológicas, Núcleo de Pesquisas em Ciências Biológicas, Laboratório de Imunopatologia, Ouro Preto, MG, Brasil
³Universidade Federal de Ouro Preto, Escola de Farmácia, Departamento de Análises Clínicas, Ouro Preto, MG, Brasil
⁴Universidade Federal de Ouro Preto, Instituto de Ciências Exatas e Biológicas, Núcleo de Pesquisas em Ciências Biológicas, Laboratório de Bioquímica e Biologia Molecular, Ouro Preto, MG, Brasil
⁵Universidade Federal de Ouro Preto, Laboratório Multiusuário de Bioquímica Nutricional, Ouro Preto, MG, Brasil

**BACKGROUND** Parasitic diseases may increase the risk of metabolic abnormalities through persistent inflammation. However, the effects of a hyperglycaemic diet during *Trypanosoma cruzi* infection remain poorly understood.

**OBJECTIVES** This study aimed to investigate the metabolic, parasitological, immunological, and histological effects of a hyperglycaemic diet during acute *T. cruzi* infection in mice.

**METHODS** C57BL/6 mice were divided into four groups: non-infected with standard diet (NISD), infected with a standard diet (ISD), non-infected with hyperglycaemic diet (NIHD), and infected with hyperglycaemic diet (IHD). Animals were fed their respective diets for eight weeks prior to infection and monitored up to 30 days post-infection (DPI) for blood glucose, body mass, biochemical markers, parasitaemia, tissue alterations, and immune cell profiles.

**FINDINGS** At the time of infection, hyperglycaemic diet groups showed higher blood glucose and body mass. By 30 DPI, these animals exhibited lower glucose, increased parasitaemia, adipose tissue hypertrophy, and reduced cholesterol levels compared with controls. Infected groups showed an increased CD4+ IFN-γ+ T cells at 10 DPI, whereas macrophage expansion was observed only in ISD mice. Cardiac parasitism was higher at 30 DPI than at 10 DPI.

**MAIN CONCLUSIONS** These results show that *T. cruzi* infection affects metabolic parameters and that a hyperglycaemic diet worsens parasitological outcomes during the acute phase of infection and appears to downregulate the immune response.

Key words: Chagas disease - *Trypanosoma cruzi* - metabolic diseases

Discovered by Carlos Chagas in 1909, Chagas disease (CD), or American trypanosomiasis, is caused by the protozoan *Trypanosoma cruzi*, transmitted primarily by the haematophagous insect vector known as the triatomine or "kissing bug".[1] In addition to vector transmission, other routes of infection include congenital transmission, blood transfusion, organ transplantation, oral transmission, and laboratory accidents.[2] Epidemiologically, CD is considered one of the major neglected tropical diseases, endemic in 21 Latin American countries, with a significant global impact. It is estimated that over 7 million people are affected by the disease, with 100 million at risk of contracting it. However, less than 10% of cases are diagnosed, contributing to an alarming number of disease-related deaths.[3]

Demographic changes and lifestyle habits are contributing to the rise of non-communicable chronic diseases, such as obesity and diabetes mellitus, which have become among the leading causes of morbidity and mortality globally, highlighting the need for a comprehensive approach to CD control.[4,5,6] A hyperglycaemic diet is characterised by the consumption of foods with a high glycaemic index, resulting in high glycaemic loads. This leads to elevated insulin levels, which activate fat synthesis and increase triacylglycerol concentrations in the blood. Such a diet can cause metabolic disturbances, including increased body mass and elevated serum levels of insulin and triacylglycerol.[7] The metabolic changes induced by a hyperglycaemic diet also affect the immune response, particularly in conditions such as

Financial support: This study was funded by UFOP, FAPEMIG, CAPES, CNPq, FINEP (including the FINEP and FAPEMIG APQ-02511-22 projects).
+ Corresponding author: paula@ufop.edu.br | ⓘ https://orcid.org/0000-0001-7033-7686

**How to cite**: das Mercês AC, Marques FS, Martins BT, Moreira GJL, Roatt BM, Carneiro CM, et al. Influence of a hyperglycaemic diet on *Trypanosoma cruzi* infection in mice model. Mem Inst Oswaldo Cruz. 2025; 120: e250092.
**Handling editor:** Adeilton Alves Brandão | ⓘ https://orcid.org/0000-0001-5877-607X

obesity. The enlargement of adipocytes can lead to the release of pro-inflammatory substances, promoting the infiltration of immune cells, such as macrophages, into affected areas of adipose tissue. This inflammatory state may contribute to the progression of health conditions associated with obesity.[8,9]

The relationship between carbohydrate consumption and parasitic diseases, including CD, is an area that remains underexplored in the scientific literature. However, studies have demonstrated a link between a hyperglycaemic diet and conditions such as obesity and type 2 diabetes mellitus, which are often observed in CD patients. For example, research assessing the prevalence of metabolic syndrome in CD patients revealed high rates of obesity, diabetes mellitus, and dyslipidaemia in this population.[10]

The accumulation of body adipose tissue and elevated blood cholesterol levels are particularly relevant for CD patients, as adipose tissue can serve as a reservoir for *T. cruzi*. This reservoir contributes to an increased parasite load and can lead to insulin resistance, thereby triggering a chronic inflammatory state. Additionally, the imbalance in the regulation of pro-inflammatory and anti-inflammatory cytokines may increase the risk of tissue damage in the host.[11,12]

A study investigating the influence of a hyperglycaemic diet on *T. cruzi* infection could clarify how diet-induced metabolic changes affect the immune response and the progression of the infection. This would help to better understand the underlying mechanisms of the interaction between diet and parasitic infection, offering insights into potential therapeutic and preventive strategies. In this context, the aim of the present work was to investigate the influence of a hyperglycaemic diet on *T. cruzi* infection in a murine model.

## MATERIALS AND METHODS

*Animals and experimental groups* - Thirty-six isogenic male C57BL/6 mice, 30 days old, born at the Animal Science Centre of the Federal University of Ouro Preto (CCA - UFOP), were used.

The animals were randomly assigned to four experimental groups: non-infected animals subjected to a standard diet (NISD, n = 8), *T. cruzi*-infected animals subjected to a standard diet (ISD, n = 10), non-infected animals subjected to a hyperglycaemic diet (NIHD, n = 8), and *T. cruzi*-infected animals subjected to a hyperglycaemic diet (IHD, n = 10). Each group was further subdivided equally for evaluation at 10 and 30 days post-infection (DPI).

All experimental procedures were conducted in accordance with the ethical principles recommended by the National Council for the Control of Animal Experimentation (CONCEA) and were approved by the Animal Use Ethics Committee of the Federal University of Ouro Preto (CEUA - UFOP) under protocol no. 8981260122.

*Diet composition* - The standard diet was formulated according to the AIN-93G diet[13] with the aim of maintaining the animals' nutritional status. The hyperglycaemic diet was prepared according to the HFrD,[14] a modified version of the AIN-93 diet, designed to induce metabolic changes related to hyperglycaemic eating habits. The composition of the diets is detailed in Table I.

*Experimental strategy* - Throughout the experiment, the animals were provided with water *ad libitum*. Eight weeks prior to the day of infection, mice were subjected to their respective diets. The infection was carried out 56 days after the introduction of the diet (0 DAI). Euthanasia was performed at 10 days after infection (10 DAI) to assess a period close to the peak of parasitaemia, while euthanasia was conducted at 30 DAI to evaluate the period following parasitaemia.

*Trypanosoma cruzi infection* - The animals were inoculated intraperitoneally with $1 \times 10^4$ bloodstream trypomastigotes forms of Colombian strain of *T. cruzi*. These forms were obtained from mice previously infected with the same strain, also via intraperitoneal injection, for strain maintenance by the Morphopathology Laboratory at the UFOP, with passages performed every 15 days in *Swiss* mice.

*Survive rate* - To determine survival rate, animals were monitored daily until the day of euthanasia. Deaths were recorded and expressed as a cumulative percentage.

*Parasitaemia curve* - After the 4th day of infection mice were evaluated daily to establish the parasitaemia curve according to Brener's adapted methodology.[15] In short, five microliters of blood were taken from the caudal vein and placed on a slide. Blood trypomastigotes were counted in 50 random microscopic fields in an optical microscope. This procedure was repeated daily until no parasites were observed for five consecutive days. Results were given as parasites/0.1 mL of blood.

### TABLE I
Composition of standard and hyperglycemic diets

| Ingredients/Kg | Standard diet (g) | Hyperglycemic diet (g) |
|---|---|---|
| Corn starch | 397,49 | |
| Fructose | - | 553 |
| Casein | 200 | 140 |
| Maltodextrin | 132 | - |
| Sucrose | 100 | 12 |
| Soybean oil | 70 | 40 |
| Cellulose | 50 | 50 |
| Mineral mix[a] | 35 | 35 |
| Vitamin mix[b] | 10 | 10 |
| L-cystine | 3 | 1,8 |
| Choline bitartrate | 2,5 | 2,5 |
| Lard | - | 154,1 |
| Tert-butylhydroquinone | 14 | - |
| BHT preservative | 2 | 2 |

a: mineral mix (g/kg): NaCl - 139.3; KI - 0.79; MgSO4.7H2O - 57.3; CaCO3 - 381.4; MnSO4.H2O - 4.01; FeSO4.7H2O - 0.548; CuSO4.5H2O - 0.477; CoCl2.6H2O - 0.023; KH2PO4 - 389.0; b: vitamin mix (mg/kg): retinol acetate - 690; cholecalciferol - 5; P-aminobenzoic acid - 10,000; inositol - 10,000; niacin - 4,000; riboflavin - 800; thiamine HCl - 500; folic acid - 200; biotin - 40; cyanocobalamin - 3; dl-α-tocopherol - 6,700; sucrose - q.s.p. 1,000.

*Ingestion quantification* - The animals' food intake was measured weekly. At the start of the diet, 105 g of food was provided. After each week, the remaining food was weighed, and the amount was replenished to 105 g using an SF-400 digital scale.

*Body mass quantification* - The animals were weighed before the start of the diet and on the day of euthanasia (10 and 30 DPI) using an SF-400 digital scale.

*Euthanasia and samples* - Four animals from the non-infected groups and five from the infected groups were euthanised at 10 and 30 DPI. Euthanasia was performed under general anaesthesia, with the administration of ketamine (90 mg/kg) and xylazine hydrochloride (9 mg/kg), followed by exsanguination, which was carried out after confirming the absence of reflexes and resistance to painful stimuli in the mice.

At euthanasia, 200 µL of blood was collected from the brachial plexus and centrifuged at 10,000 rpm for 10 min at 4ºC. The serum was then collected for biochemical analyses.

*Tissue collection* - At necropsy, fragments of the heart and adipose tissue were collected for histological analysis and quantification of tissue parasitism. All retroperitoneal and epididymal adipose tissue was removed and weighed. The spleen was collected for cellular phenotypic characterisation. The tibia was collected, cleaned, and measured using a caliper to normalise adipose tissue weight to tibia length.

*Biochemical analyses* - Serum insulin levels were quantified using the Rat/Mouse Insulin ELISA EZ-RMI-13K kit (Sigma-Aldrich), following the manufacturer's instructions. Cholesterol was measured using the Monoreagent Cholesterol kit (Bioclin, K083), following the manufacturer's recommendations, with the following adaptation for the animal model: 300 µL of Reagent No. 1. Triglycerides were measured using the Monoreagent Triglycerides kit (Bioclin, K117), with the following adaptation: 300 µL of Reagent No. 1. Capillary blood glucose was measured from the tail vein of mice fasted for 12 hours using a glucometer (On Call Plus II) at two time points: prior to the introduction of the diet and on the day of euthanasia.

*Spleen immunophenotyping* - After spleen collection, the samples were mechanically dissociated in incomplete RPMI medium using a scalpel to obtain a single-cell suspension. Subsequent steps — including cell concentration adjustment, viability staining, surface antibody staining, red blood cell lysis, intracellular cytokine staining (Table II), fixation, and storage — were performed as previously described by Vieira.[16]

A minimum of 100,000 events per sample were acquired using an LSR Fortessa flow cytometer (BD Biosciences), operated using BD FACSDiva™ software. Instrument compensation was performed using BD™ CompBeads. Dead or apoptotic cells were excluded based on FVS450 staining, which binds to surface and intracellular amines. Data were analysed using FlowJo™ v10.8.0 software (BD Biosciences). The gating strategy for characterising total lymphocytes and their subpopulations (CD4+, CD8+, and CD11b+) producing interferon (IFN)-γ, tumour necrosis factor (TNF), and interleukin (IL)-10 is shown in Fig. 1.

*Parasite load quantification* - Heart and adipose tissue fragments, previously stored at -80ºC, were dissected with a scalpel and weighed to approximately 30 mg. Samples were transferred to 1.5 mL microtubes, and DNA extraction was performed using the Wizard® Genomic DNA Purification Kit (Promega, Madison, WI, USA), following the manufacturer's instructions with minor modifications, based on the protocol described by Marques et al.[17] DNA concentration and purity were assessed using a NanoDrop 2000 spectrophotometer (Thermo Scientific, USA), and samples were stored at -20ºC until quantitative polymerase chain reaction (qPCR) analysis.

*Histopathological evaluation* - At necropsy, portions of the heart and adipose tissue were fixed in methanol/dimethylsulfoxide (80/20) and stored at 4ºC, with the fixation solution replaced daily for three consecutive days. The tissue fragments were then routinely dehydrated through an ascending series of alcohols and cleared in xylene. Subsequently, the tissues were embedded in paraffin blocks and sectioned at 4 µm thickness for slide preparation. Afterward, slides were stained with haematoxylin and eosin (H&E), dried in an oven at approxi-

TABLE II

Panel of monoclonal antibodies used in cell immunophenotyping assays and intracellular cytokine staining in the *ex vivo* context

| Marker* | Fluorochrome | Clone | Dilution | Function |
|---|---|---|---|---|
| CD3e | BV650 | 17A2 | 1:100 | Defines T lymphocytes |
| CD4 | BV605 | RM4-5 | 1:200 | Defines subpopulation of helper T lymphocytes |
| CD8a | BV786 | 53-6.7 | 1:100 | Defines subpopulation of cytotoxic T lymphocytes |
| CD11b | FITC | M1-70 | 1:80 | Defines macrophage subpopulation |
| TNF | PE-Cy7 | LG.3A10 | 1:100 | Beltcin type 1 |
| IFN-γ | PE | XMGI1.2 | 1:50 | Beltcin type 1 |
| IL-10 | BV711 | JES5-16E3 | 1:50 | Immunomodulatory beltcin |

*all markers used were from the BD bioscience brand.

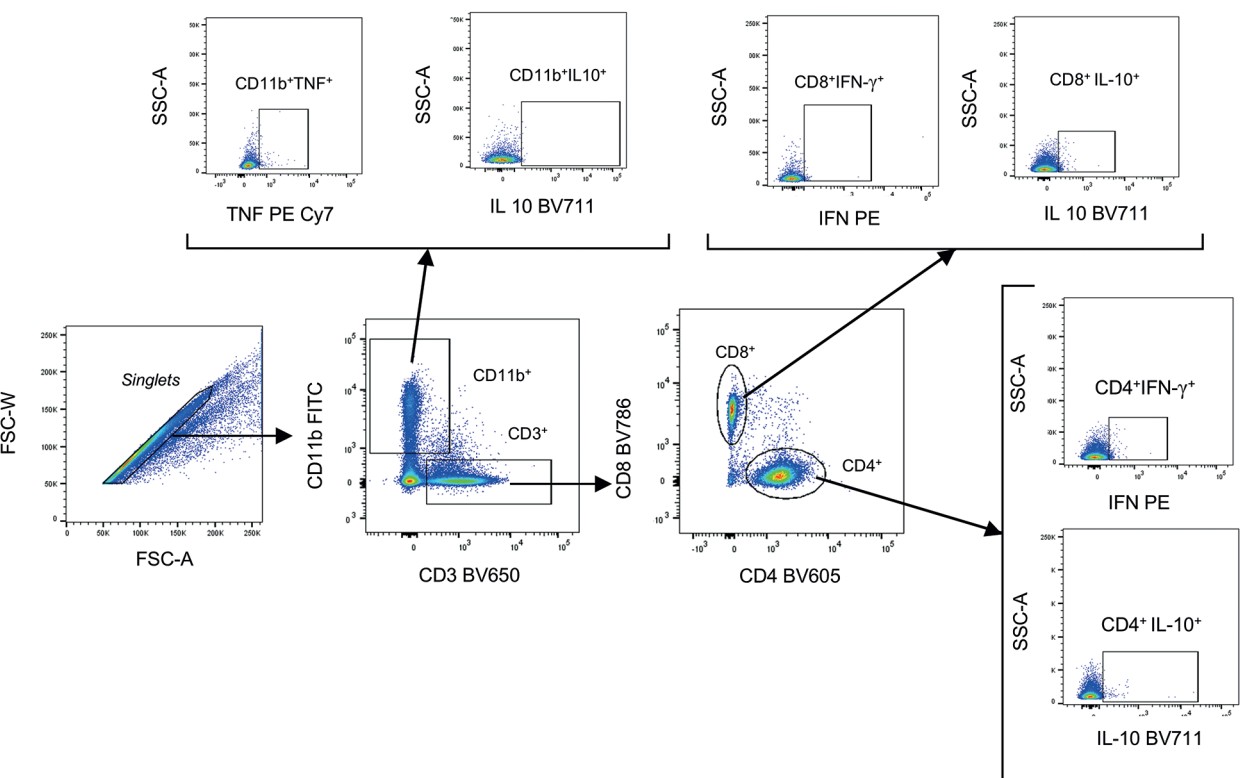

Fig. 1: proposed analysis strategy for the phenotypic characterisation of the percentage of total lymphocyte populations and their subpopulations (CD4+, CD8+, and CD11b+) producing the cytokines IFN-γ, TNF, and IL-10 from *ex vivo* spleen samples.

mately 50ºC, and once completely dried, coverslips were mounted using Entellan®.

*Morphometric analysis* - Inflammation was quantified in the heart, a target organ for the strain employed, and in adipose tissue, which has been described as a parasitic reservoir and is directly influenced by insulin. A semi-automated digital microscope (Leica DM5000B) equipped with a 5 MegaPixel CCD camera, model MC170HD, was used to capture images of the slides at 40× magnification. For the heart, 25 random fields were imaged, while for adipose tissue, 15 random fields were captured.

Following image acquisition, the cellular nuclei were quantified with Leica Qwin v3.5.1 software.

Both the microscope and the analysis software are part of the Advanced Microscopy and Microanalysis Multi-User Laboratory (LMU-MAM) at the Biological Sciences Research Centre (NUPEB), UFOP.

*Statistical analysis* - Statistical tests were performed using GraphPad Prism 8.0.2 software (Prism Software, Irvine, CA, USA). The Shapiro-Wilk test was conducted to confirm normality. For parametric data, comparisons between groups were performed using Student's t-test or one-way or two-way analysis of variance (ANOVA). When significant differences were detected, Tukey's or Bonferroni's test was applied to determine specific differences between means. For non-parametric data, the Mann-Whitney test was conducted. Differences between means were considered significant at $p < 0.05$.

## RESULTS

*Impact of diet on disease progression and tissue pathology in the experimental model* - The higher parasitaemia observed in the IHD group on days 25 and 26 (Fig. 2A) may be associated with the reduced survival in this group (Fig. 2B). While all non-infected animals survived, infected animals on the hyperglycaemic diet (IHD) exhibited lower survival (80%) compared with those on the standard diet (90%), suggesting that the increased parasite burden may contribute to disease severity and mortality.

The impact of disease progression was also evaluated by quantifying parasitism in the heart and adipose tissue. In the heart, consumption of the hyperglycaemic diet appeared to result in a higher parasite burden at 10 days after infection, which was the opposite profile to that observed in the IHD group, where a significant increase was only detected at 30 days after infection (Fig. 2C). Morphometric analysis of cardiac inflammation further revealed that, at 30 DAI, the infected group on a hyperglycaemic diet (IHD 30 DAI) exhibited a more pronounced inflammatory response compared with the non-infected group on the same diet (NIHD 30 DAI) (Fig. 2E). This indicates that the inflammatory process in the heart is associated with disease progression and the earlier increase in parasite burden.

In adipose tissue, group differences underscored a dietary influence. Higher parasite levels were observed in the IHD group at 10 DAI compared with the IHD

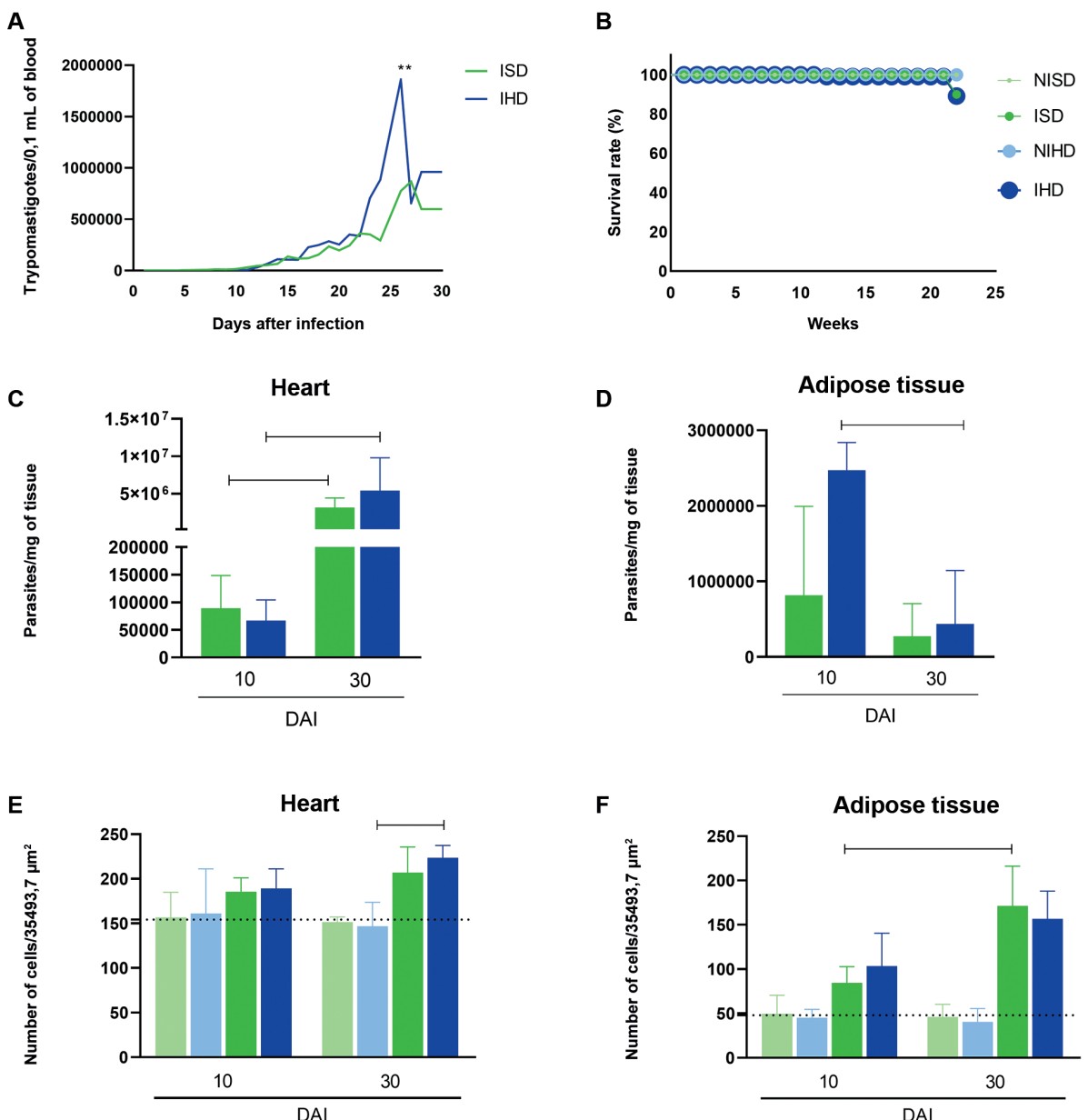

Fig. 2: impact of infection on C57BL/6 mice, either non-infected or infected with the Colombian strain of *Trypanosoma cruzi*, and subjected to a standard diet (SD) or a hyperglycaemic diet (HD) in the following groups: non-infected animals subjected to a standard diet (NISD), *T. cruzi*-infected animals subjected to a standard diet (ISD), non-infected animals subjected to a hyperglycaemic diet (NIHD), and *T. cruzi*-infected animals subjected to a hyperglycaemic diet (IHD). (A) Parasitaemia. Each curve represents the mean of 20 animals from each infected group over 30 days after infection (DAI). (B) Survival rate. Each curve represents the mean of 36 animals from each non-infected or infected group over 30 days post-infection. (C) Number of parasites in the heart. The "bar" indicates a difference between infected groups at 30 DAI compared with the same groups at 10 DAI. (D) Number of parasites in adipose tissue. The "bar" represents a decrease in tissue parasitism in the IHD group at 30 DAI compared with 10 DAI. (E) Quantification of the inflammatory process in the heart. The "bar" denotes an increase in inflammation in the infected group subjected to a hyperglycaemic diet at 30 DAI compared with its respective control. (F) Quantification of the inflammatory process in adipose tissue. The "bar" represents an increase in inflammation in the group subjected to a standard diet at 30 DAI compared with the same group at 10 DAI. Values are expressed as mean ± standard deviation.

group at 30 DAI (Fig. 2D). However, unlike the findings in the heart, the higher parasite burden at day 10 did not influence the inflammatory response, as only the ISD group displayed a significant increase at 30 DAI (Fig. 2F). These findings suggest that hyperglycaemia may modulate parasite persistence and distribution, with subsequent effects on tissue-specific inflammation in the heart, which correlates with the observed parasitaemia and survival rates.

*Impact of diet and T. cruzi infection on food intake, body composition, and metabolic parameters in mice* - To assess the effects of diet and infection, mice were moni-

tored weekly for food intake, body mass, retroperitoneal and epididymal adipose tissue mass, blood glucose, serum insulin, triacylglycerol, and total cholesterol (Fig. 3).

The analysis of weekly food intake revealed that infected animals, irrespective of diet, exhibited a significant reduction in consumption at 30 DAI compared with their respective controls. This indicates that long-term infection itself influenced feeding behaviour. Furthermore, at both 10 and 30 DAI, diet also affected intake, with the IHD group consuming less food than the ISD group (Fig. 3A). These findings are consistent with body mass measurements, as non-infected mice on the hyperglycaemic diet displayed higher body mass both at the time of infection and at 10 DAI (Fig. 3B).

The analysis of retroperitoneal adipose tissue further highlighted the impact of infection, potentially linked to food intake, as infected groups showed higher retroperitoneal adipose tissue at 10 DAI relative to 30 DAI (Fig. 3C). Diet also significantly influenced this parameter, with non-infected mice on the hyperglycaemic diet displaying higher retroperitoneal adipose tissue at both 10 and 30 DAI. Similarly, diet appeared to affect epididymal adipose tissue accumulation, as non-infected mice on the hyperglycaemic diet showed increased epididymal adipose tissue at 30 DAI compared with 10 DAI (Fig. 3D).

The effects of food intake and long-term infection were also evident in biochemical analyses. Before infection, mice fed a hyperglycaemic diet exhibited higher blood glucose levels than those on a standard diet. By 30 DAI, however, this trend had reversed, with infected animals on the standard diet showing altered glucose levels (Fig. 3E). At 30 DAI, cholesterol levels were higher in non-infected mice on a hyperglycaemic diet than in infected animals, indicating that infection distinctly affected cholesterol metabolism (Fig. 3F). No significant differences were observed in triacylglycerol levels (Fig. 3G) or serum insulin (Fig. 3H) across the experimental groups.

*Immunophenotyping of splenic mononuclear cells* - On day 10 post-infection, there were no significant differences in the percentage of IL-10-producing CD4+ (Fig. 4A) or CD8+ T lymphocytes (Fig. 4C), nor in the percentage of IFN-γ-producing CD8+ T lymphocytes (Fig. 4D). The percentage of IFN-γ-producing CD4+ T lymphocytes was higher in both infected groups (ISD and IHD) when compared with their respective non-infected control groups (NISD and NIHD) (Fig. 4B).

Regarding IL-10-producing macrophages, the percentage was higher in the ISD group compared with the NISD group (Fig. 4E). However, no significant differences were observed in the percentage of TNF-producing macrophages between the groups (Fig. 4F).

On day 30 post-infection, the percentages of IL-10-producing CD4+ and CD8+ T lymphocytes (Fig. 5A, C), as well as IFN-γ-producing CD8+ T lymphocytes (Fig. 5D), did not show statistically significant differences between the groups. However, the percentage of IFN-γ-producing CD4+ T lymphocytes was higher in the ISD group compared with the NISD group (Fig. 5B).

Additionally, the percentage of IL-10-producing macrophages was smaller in the ISD group compared with its

control (NISD) (Fig. 5E). In contrast, no significant differences were observed in the percentages of CD11+ macrophages producing TNF among the groups (Fig. 5F).

## DISCUSSION

Since CD remains a globally neglected condition with significant public health impacts, and metabolic disorders are highly prevalent in Brazil, conducting a study that explores the effects of a hyperglycaemic diet on *T. cruzi* infection in an experimental model may be useful for understanding the potential interactions between metabolic alterations and parasitic infections.

In our study, mice fed a hyperglycaemic diet exhibited higher glycaemic levels at the early time point (0 DAI), in agreement with a previous study.[18] However, at 30 DPI (DAI), infected animals under a hyperglycaemic diet showed reduced blood glucose levels, likely related to infection-associated symptoms such as lethargy and reduced food intake. Combs et al.[19] previously observed that mice infected with *T. cruzi* and subjected to a high-glucose diet showed hypoglycaemia, suggesting this may result from cytokine storms that suppress appetite and/or increase glucose uptake by the parasite.[11,20] Although amastigotes preferentially utilise amino acids and fatty acids as carbon sources, *T. cruzi* can also modulate host cell metabolism to facilitate glucose uptake, as demonstrated by Shah-Simpson.[21]

Hyperglycaemic diet intake also led to increased body weight and adipose tissue mass, particularly in non-infected groups. These findings are consistent with previous studies using carbohydrate-rich diets in rodents.[22,23] In infected groups, adipose tissue mass was reduced at 30 DAI, probably due to lower food intake. Despite known associations between hyperglycaemia and increased cholesterol levels, infected animals under the hyperglycaemic diet showed reduced total cholesterol, potentially reflecting the impact of parasitaemia on appetite and lipid metabolism. Inflammation and infection are known to induce marked changes in lipid profiles, particularly reductions in total cholesterol, LDL-C, and HDL-C, as part of the host defence response.[24,25]

Regarding the immune response, higher percentages of IFN-γ-producing CD4+ T lymphocytes were observed at 10 and 30 DAI, consistent with the inflammatory profile expected during acute *T. cruzi* infection.[26,27] Interestingly, non-infected mice under a hyperglycaemic diet also showed increased IFN-γ expression, possibly due to diet-induced inflammation. This aligns with reports that obesity enhances IFN-γ production and the antigen presentation capacity of adipocytes.[28,29]

Macrophages are central to early immune responses against *T. cruzi*, particularly via M2 polarisation which enhances phagocytosis and parasite clearance.[30,31] In our study, infected animals under a standard diet showed increased macrophage populations, suggesting a host attempt to control parasitaemia.

Cardiac parasitic load was higher at 30 DAI compared with 10 DAI, particularly in animals infected with the Colombian strain of *T. cruzi*, known for its cardiac tropism.[32] Chronic parasitism in the heart promotes per-

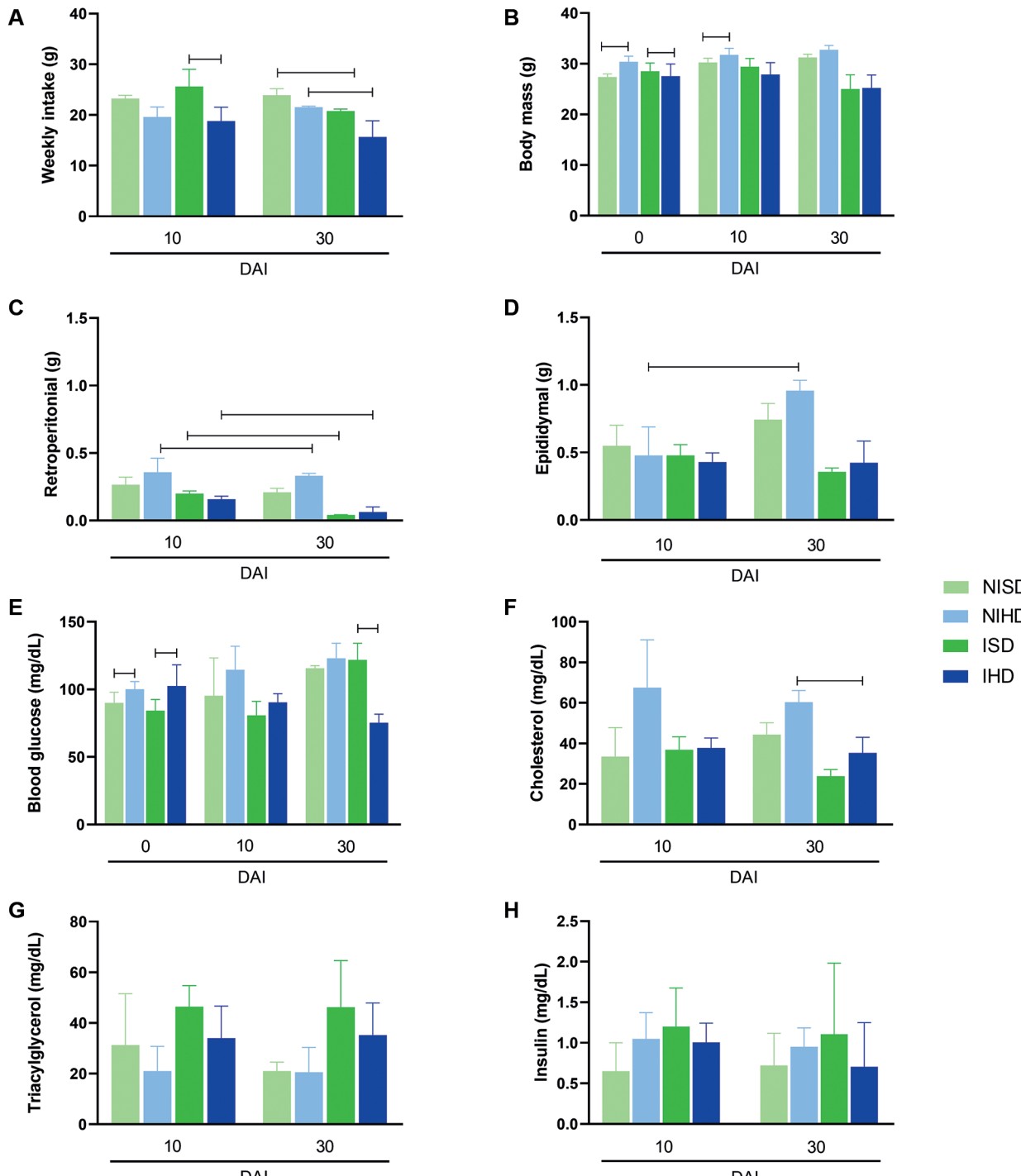

Fig. 3: impact of infection and/or diet on food consumption, body weight, and metabolic profile in C57BL/6 mice, either non-infected or infected with the Colombian strain of *Trypanosoma cruzi*, and subjected to a standard diet (SD) or a hyperglycaemic diet (HD) in the following groups: non-infected animals subjected to a standard diet (NISD), *T. cruzi*-infected animals subjected to a standard diet (ISD), non-infected animals subjected to a hyperglycaemic diet (NIHD), and *T. cruzi*-infected animals subjected to a hyperglycaemic diet (IHD). (A) Food intake quantification. The "bar" indicates statistically significant differences between infected groups at 10 days after infection (DAI) and between infected groups and their respective controls at 30 DAI (p < 0.05). (B) Body mass. The "bar" denotes statistically significant differences between non-infected and infected groups prior to infection and between non-infected groups at 10 DAI. (C) Retroperitoneal adipose tissue mass. The "bar" indicates a decrease in retroperitoneal adipose tissue content in the NIHD group at 30 DAI compared to 10 DAI, as well as a decrease in both infected groups at 30 DAI compared to 10 DAI. (D) Epididymal adipose tissue mass. The "bar" represents an increase in epididymal adipose tissue content in the NIHD group at 30 DAI compared to 10 DAI. (E) Blood glucose quantification. The "bar" denotes higher glucose levels in mice subjected to a hyperglycaemic diet compared to those on a standard diet prior to infection, and a decrease in glucose levels in the IHD group compared to the ISD group at 30 DAI. (F) Serum cholesterol levels. The "bar" represents a reduction in cholesterol levels in infected mice subjected to a hyperglycaemic diet compared to their respective controls, as well as a difference between the NIHD and IHD groups at 30 DAI. (G) Triacylglycerol levels. (H) Serum insulin levels. Values are expressed as mean ± standard deviation.

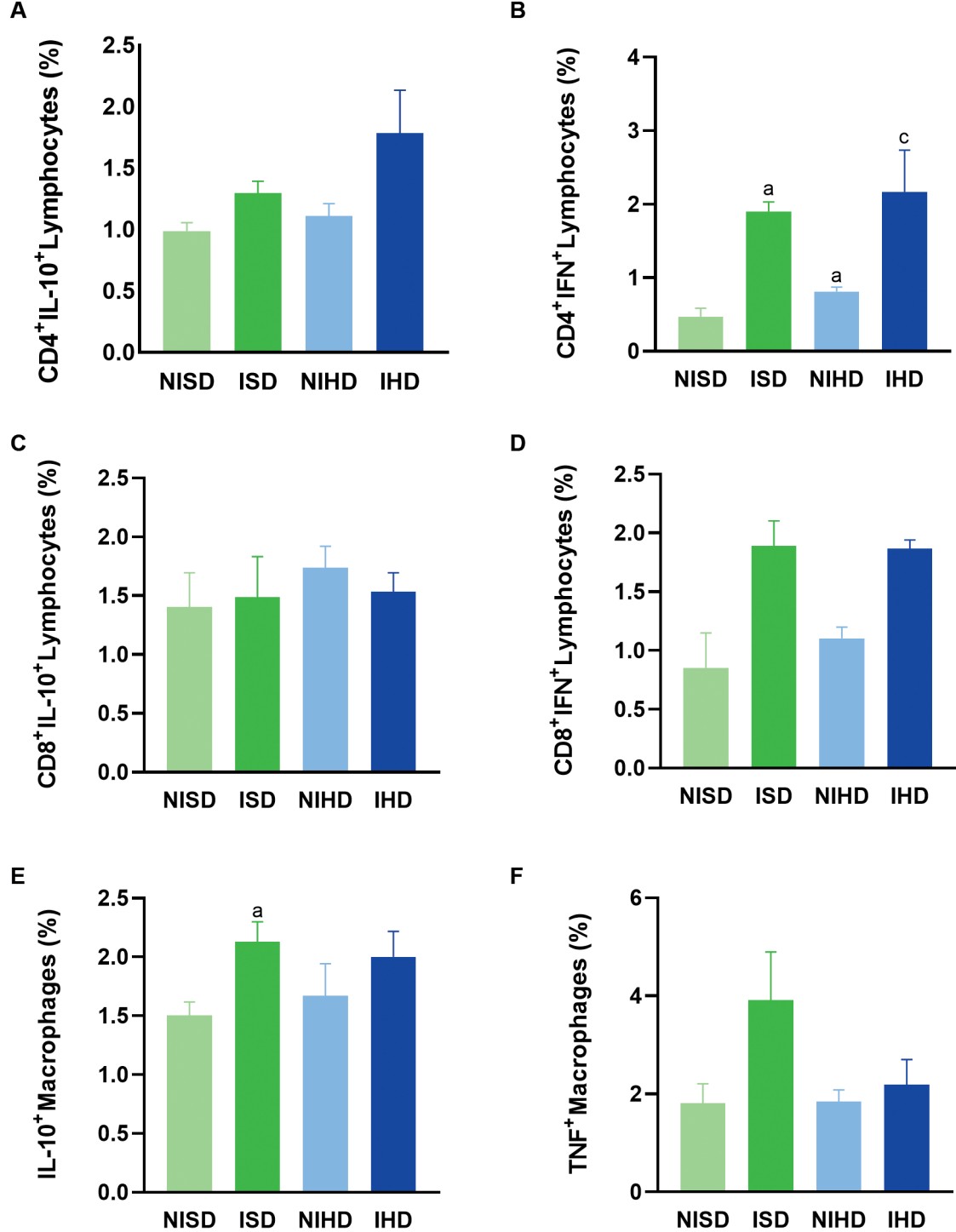

Fig. 4: immunophenotyping of splenic mononuclear cells in the following groups: non-infected animals subjected to a standard diet (NISD), *Trypanosoma cruzi*-infected animals subjected to a standard diet (ISD), non-infected animals subjected to a hyperglycaemic diet (NIHD), and *T. cruzi*-infected animals subjected to a hyperglycaemic diet (IHD). (A) Percentage of CD4+ IL-10+ T lymphocytes; no statistically significant differences were observed between the groups. (B) Percentage of CD4+ IFN-γ+ T lymphocytes; the letters "a" and "c" indicate statistically significant differences compared with the control group (NISD) and the non-infected group subjected to the hyperglycaemic diet (NIHD), respectively. (C) Percentage of CD8+ IL-10+ T lymphocytes; no statistically significant differences were observed between the groups. (D) Percentage of CD8+ IFN-γ+ T lymphocytes; no statistically significant differences were observed between the groups. (E) Percentage of IL-10+ macrophages; the letter "a" denotes a statistically significant difference compared with the control group (NISD). (F) Percentage of TNF+ macrophages; no statistically significant differences were observed between the groups. Values are expressed as mean ± standard deviation.

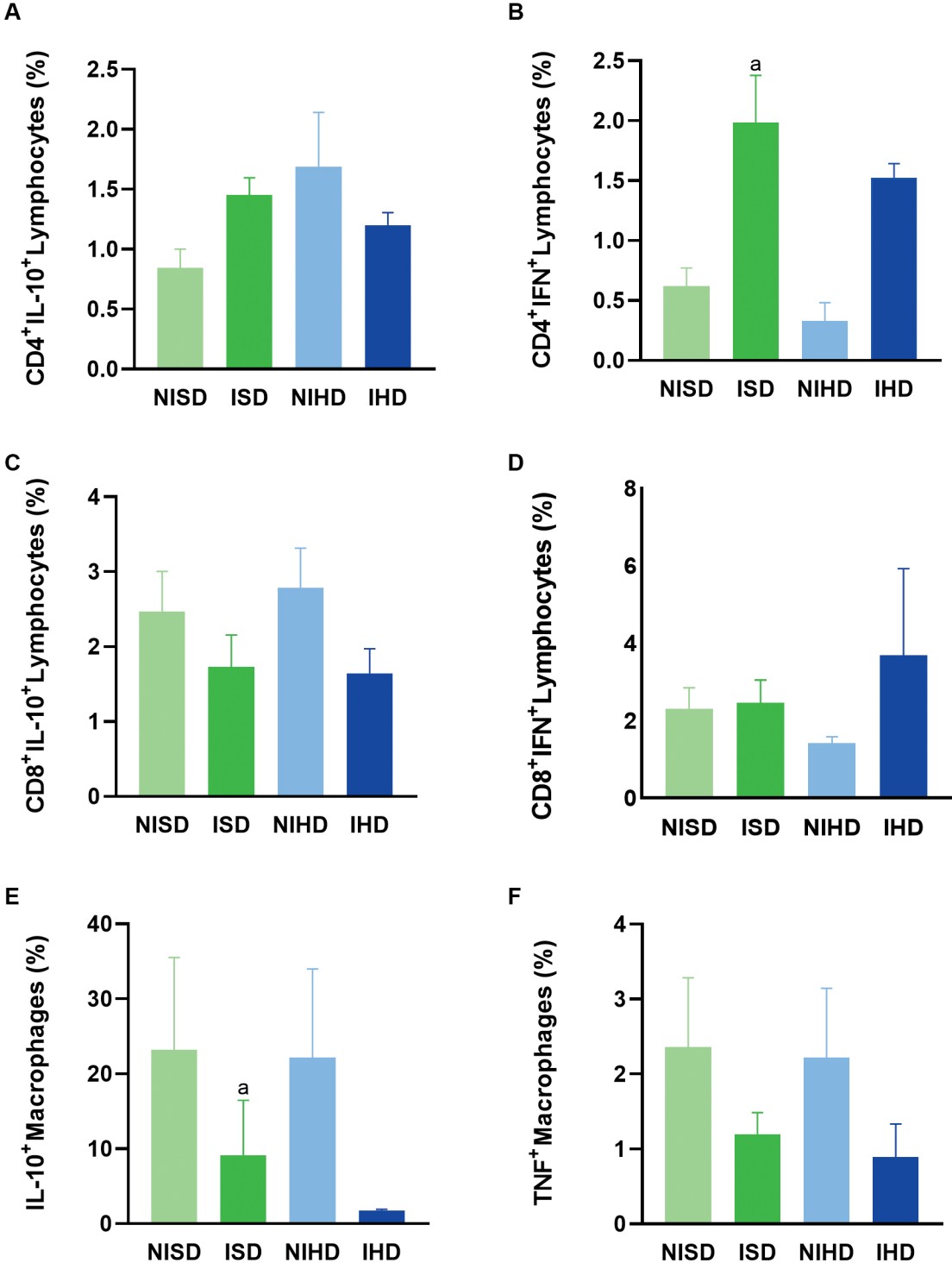

Fig. 5: phenotyping of splenic mononuclear cell profiles in C57BL/6 mice in the following groups: non-infected animals subjected to a standard diet (NISD), *Trypanosoma cruzi*-infected animals subjected to a standard diet (ISD), non-infected animals subjected to a hyperglycaemic diet (NIHD), and *T. cruzi*-infected animals subjected to a hyperglycaemic diet (IHD). (A) Percentage of CD4+ IL-10+ T lymphocytes; no statistically significant differences were observed between the groups. (B) Percentage of CD4+ IFN-γ+ T lymphocytes; the letter "a" indicates a significant difference from the control group (NISD). (C) Percentage of CD8+ IL-10+ T lymphocytes; no statistically significant differences were observed between the groups. (D) Percentage of CD8+ IFN-γ+ T lymphocytes; no statistically significant differences were observed between the groups. (E) Percentage of IL-10+ macrophages; no statistically significant differences were observed between the groups. (F) Percentage of TNF+ macrophages; no statistically significant differences were observed between the groups. Values are expressed as mean ± standard deviation.

sistent inflammation, necrosis, and fibrosis, ultimately leading to progressive myocardial damage, as previously demonstrated in both human and murine studies.[33,34,35]

Adipose tissue also emerged as a relevant site of parasitic persistence. In our study, mice on a hyperglycaemic diet showed higher adipose tissue parasitic loads at day 10 compared with day 30 post-infection. It was demonstrated in a murine model that *T. cruzi* infects adipose tissue early and remains detectable into the chronic phase. Moreover, DNA from *T. cruzi* was identified by polymerase chain reaction (PCR) in adipose tissue from three out of ten seropositive patients with cardiomyopathy and conduction defects.[36] Likewise, found viable parasites in adipose tissue even after 300 days of infection, with parasite loads comparable to those in the heart. This persistence may be explained by the slow turnover of adipocytes, which are long-lived and metabolically active cells.[19]

The ability of *T. cruzi* to persist in adipose tissue has important implications for treatment. The parasite's presence in a low-turnover, poorly vascularised tissue may reduce the efficacy of benznidazole and facilitate immune evasion. Such characteristics are not exclusive to CD; adipose tissue has been shown to serve as a reservoir in other infections, including Brill-Zinsser disease,[37] tuberculosis,[38] and malaria.[39]

In the context of HIV, both the infection and antiretroviral therapy may induce lipodystrophy, characterised by reduced subcutaneous fat and redistribution of fat to central depots. This remodelling of adipose tissue could lead to the release of intracellular *T. cruzi* into circulation, favouring disease reactivation in immunocompromised patients.[40]

Together, our findings reinforce the role of adipose tissue as an active participant in the immunometabolic alterations observed during *T. cruzi* infection. Its function as a parasitic reservoir and immune modulator underscores the need to consider metabolic status and tissue-specific parasite dynamics when evaluating treatment efficacy and disease progression.

The results of this study, highlighting adipose tissue as a potential reservoir for *T. cruzi*, the high cardiac parasite load during the late acute phase, and the occurrence of hypoglycaemia in infected animals, underscore the complex interaction between metabolic and infectious processes. Extrapolating from the murine model to human disease, these findings emphasise the need for more comprehensive clinical monitoring of Chagas patients, particularly those with metabolic comorbidities. Regular assessment of lipid profiles, glycaemic status, and muscle mass may be essential for improved management and prognosis in chronic CD, especially in the context of altered metabolism and potential therapeutic failure.

*In conclusion* - The results indicate that infection with *T. cruzi* was either directly or indirectly responsible for the observed metabolic alterations, and that consumption of a hyperglycaemic diet significantly influenced parasitological parameters during the acute phase of infection.

## ACKNOWLEDGEMENTS

To the Laboratório Multiusuários de Microscopia Avançada e Microanálise and the Laboratório de Citometria de Fluxo, Núcleo de Pesquisas em Ciências Biológicas at the Universidade Federal de Ouro Preto. We also thank the Animal Science Centre for providing the animals and the infrastructure necessary for their maintenance.

## AUTHORS' CONTRIBUTION

ACM - methodology, investigation, formal analysis, writing - original draft; FSM - methodology, data curation, writing - review & editing; GJLM - performed flow cytometry, data collection, and analysis; BTM - statistical analyses; SPG - biochemical analyses; JFA - diet preparation; CMC - methodology; BMR - statistical analysis of flow cytometry data; PMAV - conceptualisation, resources, writing - review & editing, supervision, project administration, funding acquisition. The authors declare no competing interests.

## DATA AVAILABILITY

The contents underlying the research text are included in the manuscript.

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

# OPEN PEER REVIEW

Memórias do IOC thanks the anonymous reviewers for their contribution to the peer review of this work.

## FIRST REVIEW ROUND

**REVIEWERS COMMENTS**

### REVIEWER #1

The manuscript is extremely relevant because it demonstrates the impact of a hyperglycemic diet on T. cruzi infection, indicating that it mainly alters parasitological parameters and the regulation of the immune response.

Some modifications should be made to improve the understanding of the manuscript

a) Introduction

1. In the introduction section, line 71, insert a more current reference of epidemiological data.

2. In the introduction section, lines 87-89, "However, studies have demonstrated a link between a hyperglycemic diet and conditions such as obesity and type 2 diabetes mellitus, which are often observed in CD patients". Insert bibliographic references related to these studies.

3. In the introduction section, lines 100-102, "This would help to better understand the underlying mechanisms of the interaction between diet and parasitic infection, offering insights into potential therapeutic and preventive strategies". What would these potential therapeutic and preventive strategies be based on studies of the mechanisms of interaction between diet and parasitic infection?

b) Methods

1. Why male C57BL/6 mice were used in this study?

2. In the methods section, lines 113 and 115, "groups: non-infected subjected to a standard diet (NISD) (n=8), infected subjected to a standard diet (ISD) (n=10), non-infected subjected to a hyperglycemic diet (NIHD) (n=8), and infected subjected to a hyperglycemic and infected subjected to a hyperglycemic diet (IHD) (n=10), with half of each group divided for 10 and 30 DPI (days post-infection)." Mention that ISD and NIHD groups were infected by T. cruzi. Was a sample calculation of these groups of animals carried out?

3. Why was the infection performed with the Colombian strain of T. cruzi? The authors believe that the genetic group or DTU of the parasite could interfere with the results of the study?

4. Why was FC receptor blocking not used prior to surface antibody labeling to prevent nonspecific binding?

5. Why did the authors not evaluate the B lymphocyte profile in the spleen?

c) Results

1. In Figure 2B it was not possible to observe the curve referring to the survival rate of the NIHD group.

2. Describe the results obtained in Figures 4A, 4C and 4D even if no statistical differences were observed.

3. In the results section, lines 338 and 339, "Additionally, the percentage of IL-10-producing macrophages was greater in the ISD group compared to its control (NISD) (Figure 5-E)." The sentence is incorrect. The percentage of IL-10-producing macrophages was smaller in the ISD group compared to its control (NISD).

4. In the Figures caption please change the symbol and write the word "bar" as it gives the impression that it is the letter "H".

5. In the Figure 2C, please change the scale of the graph as it is not possible to verify the groups with low parasite load.

6. Please correct the caption of the Figure 3C. Replace the NISD group with NIHD group.

7. Please correct the caption of the Figure 3F. Replace the NIDH with NIHD and IDH with IHD.

8. In Figure 4 the group acronyms are written incorrectly. Please correct them.

9. In Figure 5 the group acronyms are written incorrectly. Please correct them.

10. In the manuscript text it is written that CD11c was used and in the Table 1 was used CD11b. Would it be CD11b or CD11c?

c) Discussion

1. In the discussion section, lines 343-346. Authors should be careful with these sentences "focusing on its prevalence in Brazil" and "It emphasizes the need to address Chagas disease, a globally neglected condition with significant health impacts", since the work was carried out on an experimental model and not on humans.

2. In the discussion section, lines 354-358, "While prolonged hyperglycemic diets were expected to induce insulin resistance and hyperglycemia (Jung and Choi, 2017), infection symptoms such as lethargy and reduced appetite likely caused this decrease. Lower food intake and higher parasitemia in the IHD 30 DAI group contributed to the reduced glucose levels." What is the authors' hypothesis for the group with IHD having lower food intake that justifies the lower blood glucose levels than those who received the standard diet?

3. In the discussion section, lines 368-370. The authors wrote that "to increased adipose tissue mass, particularly retroperitoneal fat, as observed in non-infected groups (NIHD 10 DAI and NIHD 30 DAI) compared to controls (NISD 10 DIA and NISD 30 DIA)", but if this sentence refers to the result in Figure 3C, the difference in relation to the NISD group is not demonstrated in the graph.

4. In the discussion section, lines 384-385, "Regarding total serum cholesterol, infected animals on the hyperglycemic diet for 30 DIA (NIHD 30 DAI) […]" Replace the NIHD group with the IHD group.

5. In the discussion section, line 386, "[…] levels compared to non-infected animals on the same diet for the same period (IHD 30 DAI). Replace the IHD group with the NIHD group.

6. In the discussion section, lines 388 to 390, "This may explain the lower cholesterol levels in the NIHD 30 DAI group compared to the IHD 30 DAI group, as the NIHD 30 DAI group had a lower weekly intake of the diet." The sentence is incorrect. Figure 3F demonstrates higher cholesterol levels in the NIHD 30 DAI group compared to the IHD 30 DAI group. Furthermore, Figure 3A demonstrated that the NIHD 30 DAI group had a higher food intake than the IHD 30 DAI group.

7. Days 86 and 66 refer to 10 DAI and 30 DAI, respectively? This data is not shown in the figures.

8. In the discussion section, lines 412, "Parasitic loads were higher in the heart on day 86 compared to day 66 in both infected groups, likely due to the heart tropism of the Colombian strain.". If this sentence is referring to Figure 3C, it is incorrect, as the IHD group presented a higher parasite load on the 10 DAI and the ISD group on the 30 DAI in the heart.

9. In the discussion section, lines 419 and 420, "[…] with no differences in other groups or liver tissue." No analysis was performed on the liver. The sentence is incorrect.

10. What are the perspectives of the study? The authors intend to perform biochemical, parasitological and immune response analyses in mice in the chronic phase, that is, with a longer time of T. cruzi infection?

## REVIEWER #2

Influence of a Hyperglycemic Diet on Trypanosoma cruzi Infection in Mice Model

a) Adequacy of the abstract;

The summary is well structured and provides basic and fundamental information about the content that will be covered in the article.

- In the summary, I suggest changing the information on lines 27 and 28: "However, the impact of a hyperglycemic diet during T. cruzi infection has not been explored in the literature" by "However, the impact of a hyperglycemic diet during T. cruzi infection has been little explored in the literature".

b) Originality and importance of the contribution for the development of the field of study. And relevance of:

As mentioned in the article itself, issues involving metabolic disorders and Chagas disease have been little explored in the literature, which makes clear the importance and relevance of this study in the current scenario.

c) Introduction:

- In the introduction, I suggest changing the reference in line 66 (Nunes et al.2018) to WHO (https://dndial.org/doencas/doenca-de-chagas).. The current reference cites a study involving Chagas heart disease and would not be the best reference to talk about transmission routes.

- The global epidemiological information on Chagas disease presented in lines 68 and 69 must be updated and I suggest using the WHO data available at: (https://www.who.int/news-room/fact-sheets/detail/chagas-disease-(american-trypanosomiasis). Both DNDi and WHO currently report more than 7 million people infected with T. cruzi and more than 100 million people living in risk areas.

d) Methodology

The methodology applied to the study is well written and detailed, which allows for a good understanding of the procedures performed. It is worth highlighting the variety of techniques used. It is necessary to correct the numbering of the topics in line 164: change 2.9.2 to 2.9.1. I also suggest reducing or simplifying the methodology in items 2.11 and 2.12, adding references.

e) Results

- The Figure 1 is not cited in the text.

- In figure 2A: I suggest changing the colors of the lines by adding very distinct color.

- The Figure 2C is incomplete and it is not possible to verify what is written in the figure.

- The graphs in figure 4 have the abbreviations in the legend in English and on the X axis in Portuguese.

- Figures 4A, 4C, 4D and 4F were not cited or described in the text.

- The graphs in figure 5 have the abbreviations in the legend in English and on the X axis in Portuguese.

f) Discussion

- In line 343 replace Trypanosoma cruzi with T. cruzi.

- The discussion presented here is quite superficial and does not correlate the data found in the study. I suggest deepening the discussion, aiming to even extrapolate the observations made in the study with prognostic perspectives in humans.

g) References

- The reference Santos et al. 2018 is incomplete.

h) Figures and tables.

- The figures and tables are well presented and easy to understand. Some changes/suggestions have been made above.

**AUTHORS' RESPONSE TO THE REVIEWERS**

To: Dr. Adeilton Brandão, Ph. D.
Handling Editor
Memórias do Instituto Oswaldo Cruz
Ref: MIOC-2025-0092
Ouro Preto, June, 2025

Dear Dr. Adeilton Brandão,

We have received the e-mail containing the comments from the Editor and Reviewers regarding our manuscript (MIOC-2025-0092), entitled: "Influence of a Hyperglycemic Diet on Trypanosoma cruzi Infection in a Mouse Model" by Merces et al., submitted to Memórias do Instituto Oswaldo Cruz.

We carefully addressed all the points raised by the Editor and Reviewers, responding to each comment in detail to improve the quality of our manuscript and ensure it meets the standards for publication in Memórias do Instituto Oswaldo Cruz.

Below, we present the responses to all the queries raised by the Editor and Reviewers. The corresponding modifications in the manuscript are highlighted in green. As requested, our replies to each reviewer's comment are provided below, formatted in "Arial Black" font.

**REVIEWERS' COMMENTS**

We thank Reviewer #1 and Reviewer #2 for their careful review of our manuscript. Our responses to their questions are provided below:

**REVIEWER #1**

Introduction

1. In the introduction section, line 71, insert a more current reference of epidemiological data.

Thank you for the suggestion and we have added a more up-to-date reference.

"It is estimated that over 7 million people are affected by the disease, with 100 million at risk of contracting it. However, less than 10% of cases are diagnosed, contributing to an alarming number of disease-related deaths (WHO, 2025)."

2. In the introduction section, lines 87-89, "However, studies have demonstrated a link between a hyperglycemic diet and conditions such as obesity and type 2 diabetes mellitus, which are often observed in CD patients". Insert bibliographic references related to these studies.

"However, studies have demonstrated a link between a hyperglycemic diet and conditions such as obesity and type 2 diabetes mellitus, which are often observed in CD patients. For example, research assessing the prevalence of metabolic syndrome in CD patients revealed high rates of obesity, diabetes mellitus, and dyslipidemia in this population (Xavier et al., 2019)."

3. In the introduction section, lines 100-102, "This would help to better understand the underlying mechanisms of the interaction between diet and parasitic infection, offering insights into potential therapeutic and preventive strategies". What would these potential therapeutic and preventive strategies be based on studies of the mechanisms of interaction between diet and parasitic infection?

Since a hyperglycemic diet is associated with an inflammatory state that could exacerbate tissue damage caused by parasitic infection, the implementation of specific dietary guidelines could help attenuate disease progression in infected individuals. Given that adipose tissue serves as a reservoir for T. cruzi and plays a key role in modulating the immune response, pharmacological agents that improve metabolic control, such as insulin sensitizers and lipid-lowering drugs, could be employed as adjuncts to etiological treatment with benznidazole, particularly in Chagas disease patients with metabolic syndrome.

Furthermore, in endemic areas with a higher prevalence of individuals affected by diabetes, hypertension, and obesity, a better understanding of the correlation between metabolic syndrome and the course of T. cruzi infection may be essential for developing personalized prevention programs that integrate both disease control and improved management of metabolic syndrome.

Methods

1. Why male C57BL/6 mice were used in this study?

We chose male C57BL/6 mice because they are an isogenic strain available at our Animal Science Center, allowing us to minimize genetic variability in the study. Additionally, this strain exhibits partial resistance to Trypanosoma cruzi infection, which helps ensure that the inflammatory response generated by the infection itself does not overshadow the effects of the diet.

2. In the methods section, lines 113 and 115, "groups: non-infected subjected to a standard diet (NISD) (n=8),

infected subjected to a standard diet (ISD) (n=10), non-infected subjected to a hyperglycemic diet (NIHD) (n=8), and infected subjected to a hyperglycemic and infected subjected to a hyperglycemic diet (IHD) (n=10), with half of each group divided for 10 and 30 DPI (days post-infection)." Mention that ISD and NIHD groups were infected by T. cruzi. Was a sample calculation of these groups of animals carried out?

Only the ISD and IHD groups were infected with Trypanosoma cruzi. The NIHD group was not infected; it was only subjected to the hyperglycemic diet. We will clarify this point in the revised version of the manuscript to avoid any misunderstanding.

"The animals were randomly assigned to four experimental groups: non-infected animals subjected to a standard diet (NISD, n = 8), T. cruzi-infected animals subjected to a standard diet (ISD, n = 10), non-infected animals subjected to a hyperglycemic diet (NIHD, n = 8), and T. cruzi-infected animals subjected to a hyperglycemic diet (IHD, n = 10). Each group was further subdivided equally for evaluation at 10- and 30-days post-infection (DPI)."

The sample size was calculated using BioEstat software, version 5.3, which is widely used for statistical analyses in biological studies. The calculation considered a minimum detectable difference of 50 units between treatment means, with a standard error deviation of 20. The analysis included four treatment groups, with a statistical power of 95% and a significance level ($\alpha$) of 0.05. Additionally, potential mortality in the infected groups due to the infection, as well as natural mortality in the non-infected groups, was taken into account during the sample size estimation.

3. Why was the infection performed with the Colombian strain of T. cruzi? The authors believe that the genetic group or DTU of the parasite could interfere with the results of the study?

The Colombian strain was chosen due to its ability to induce a chronic infection in mice, which allows for the evaluation of long-term effects of a hyperglycemic diet. This strain is known to promote a clinical condition that mimics Chagas disease in murine models, including sustained inflammation, fibrosis and oxidative stress in the heart. And as we intended to evaluate the impact of the hyperglycemic diet on the course of the infection, we chose this strain due to the better survival of the animals when infected with an intermediate inoculum by this strain.

4. Why was FC receptor blocking not used prior to surface antibody labeling to prevent nonspecific binding?

In all the flow cytometry experiments carried out in this study, to avoid non-specific labeling, we diluted the antibodies in a solution containing normal serum (3-5%) from the host in which the antibody was produced. In this way, we ensured that the markings observed in the experiments were linked to the identification of the cell populations evaluated and that there were no non-specific markings. In this way, we were able to guarantee the quality of the experiments and the markings, as well as reducing the costs involved in immunophenotyping experiments.

5. Why did the authors not evaluate the B lymphocyte profile in the spleen?

In this first study, we chose to evaluate the cellular immune response, since hyperglycemia mainly affects this kind of immune response. These mechanisms include suppression of cytokine production, defects in phagocytosis, dysfunction of T cells, and leukocyte recruitment inhibition. However, in future studies we intend to expand the assessments to include other cells involved in the humoral immune response, although it is less affected by the hyperglycemic state.

Results

1. In Figure 2B it was not possible to observe the curve referring to the survival rate of the NIHD group.

We appreciate the critical analysis and have made the appropriate modifications to better visualize the data, as demonstrated in the new submitted figure 2.

2. Describe the results obtained in Figures 4A, 4C and 4D even if no statistical differences were observed.

We appreciate the critical analysis and have included the following description in the text:

"On day 10 post-infection, there were no significant differences in the percentage of IL-10-producing CD4$^+$ (Figure 4-A) or CD8$^+$ T lymphocytes (Figure 4-C), nor in the percentage of IFN-$\gamma$-producing CD8$^+$ T lymphocytes (Figure 4-D)."

3. In the results section, lines 338 and 339, "Additionally, the percentage of IL-10-producing macrophages was greater in the ISD group compared to its control (NISD) (Figure 5-E)." The sentence is incorrect. The percentage of IL-10-producing macrophages was smaller in the ISD group compared to its control (NISD).

We appreciate your critical review and have made appropriate modifications.

"Additionally, the percentage of IL-10-producing macrophages was smaller in the ISD group when compared to its control (NISD) (Figure 5-E)."

4. In the Figures caption please change the symbol and write the word "bar" as it gives the impression that it is the letter "H".

We appreciate your critical review and have made appropriate modifications in the figures caption.

5. In the Figure 2C, please change the scale of the graph as it is not possible to verify the groups with low parasite load.

We appreciate your critical review and have made appropriate modifications in the scale of the graph.

6. Please correct the caption of the Figure 3C. Replace the NISD group with NIHD group.

7. Please correct the caption of the Figure 3F. Replace the NIDH group with NIHD group and IDH with IHD.

8. In Figure 4 the group acronyms are written incorrectly. Please correct them.

9. In Figure 5 the group acronyms are written incorrectly. Please correct them.

We appreciate your critical review and have made appropriate modifications.

10. In the manuscript text it is written that CD11c was used and in the Table 1 was used CD11b. Would it be CD11b or CD11c?

We are thankful for this observation. The table contains the correct information: the marker used was CD11b. The correction has been made in the manuscript and is highlighted in green.

Discussion

1. In the discussion section, lines 343-346. Authors should be careful with these sentences "focusing on its prevalence in Brazil" and "It emphasizes the need to address Chagas disease, a globally neglected condition with significant health impacts", since the work was carried out on an experimental model and not on humans.

We agree that this sentence could lead to misunderstandings, as the study was conducted in an experimental model. Therefore, we decided to revise the entire sentence to ensure greater clarity.

2. In the discussion section, lines 354-358, "While prolonged hyperglycemic diets were expected to induce insulin resistance and hyperglycemia (Jung and Choi, 2017), infection symptoms such as lethargy and reduced appetite likely caused this decrease. Lower food intake and higher parasitemia in the IHD 30 DAI group contributed to the reduced glucose levels." What is the authors' hypothesis for the group with IHD having lower food intake that justifies the lower blood glucose levels than those who received the standard diet?

As the IHD group had higher parasitemia around 30 DAI compared to the ISD group, this may have been associated with lethargy, fever due to worsening of the parasitological condition and consequently reduced food intake, resulting in lower blood glucose levels.

3. In the discussion section, lines 368-370. The authors wrote that "to increased adipose tissue mass, particularly retroperitoneal fat, as observed in non-infected groups (NIHD 10 DAI and NIHD 30 DAI) compared to controls (NISD 10 DIA and NISD 30 DIA)", but if this sentence refers to the result in Figure 3C, the difference in relation to the NISD group is not demonstrated in the graph.

We appreciate this observation. The difference in retroperitoneal fat was observed within the NIHD group when comparing the two time points (10 and 30 DAI), and not between NIHD and NISD groups, as previously stated. We have corrected the sentence in the manuscript to accurately reflect this result shown in Figure 3C.

4. In the discussion section, lines 384-385, "Regarding total serum cholesterol, infected animals on the hyperglycemic diet for 30 DIA (NIHD 30 DAI) […]" Replace the NIHD group with the IHD group.

We appreciate the critical analysis and we have already made the correction, as highlighted in the text.

"Regarding total serum cholesterol, infected animals on the hyperglycemic diet at 30 DAI (IHD 30 DAI) had lower levels compared to non-infected animals on the same diet in the same period (NIHD 30 DAI)."

5. In the discussion section, line 386, "[…] levels compared to non-infected animals on the same diet for the same period (IHD 30 DAI). Replace the IHD group with the NIHD group.

We appreciate the critical analysis and we have already made the correction, as highlighted in the text.

"Regarding total serum cholesterol, infected animals on the hyperglycemic diet at 30 DAI (IHD 30 DAI) had lower levels compared to non-infected animals on the same diet in the same period (NIHD 30 DAI)."

6. In the discussion section, lines 388 to 390, "This may explain the lower cholesterol levels in the NIHD 30 DAI group compared to the IHD 30 DAI group, as the NIHD 30 DAI group had a lower weekly intake of the diet." The sentence is incorrect. Figure 3F demonstrates higher cholesterol levels in the NIHD 30 DAI group compared to the IHD 30 DAI group. Furthermore, Figure 3A demonstrated that the NIHD 30 DAI group had a higher food intake than the IHD 30 DAI group.

Thank you for your observation. The description of the results was inaccurate. We have revised the text to correctly reflect that, at 30 DAI, the NIHD group had higher cholesterol levels and higher food intake compared to the IHD group. We modified the description to emphasize that, although hyperglycemic diets are typically associated with increased cholesterol levels, in the presence of infection, reduced food intake likely contributed to the lower cholesterol levels observed in the IHD group. The manuscript has been updated accordingly.

7. Days 86 and 66 refer to 10 DAI and 30 DAI, respectively? This data is not shown in the figures.

Yes, days 86 and 66 correspond to 10 and 30 days after infection (DAI), respectively. The text has been revised to clarify this information and prevent any misunderstanding.

8. In the discussion section, lines 412, "Parasitic loads were higher in the heart on day 86 compared to day 66 in both infected groups, likely due to the heart tropism of the Colombian strain.". If this sentence is referring to Figure 3C, it is incorrect, as the IHD group presented a higher parasite load on the 10 DAI and the ISD group on the 30 DAI in the heart.

Thank you for your comment. As indicated in item 5 of the results, we have modified the scale of Figure 2C to improve comprehension. With this adjustment, it is clear that parasitic loads in the heart increased from 10 to 30 DAI in both infected groups, supporting the correctness of the sentence in the manuscript.

9. In the discussion section, lines 419 and 420, "[…] with no differences in other groups or liver tissue." No analysis was performed on the liver. The sentence is incorrect.

We have removed this sentence from the manuscript. Although the liver was collected and the analysis is currently ongoing, the results are not presented in this study.

10. What are the perspectives of the study? The authors intend to perform biochemical, parasitological and immune response analyses in mice in the chronic phase, that is, with a longer time of T. cruzi infection?

Yes, we intend to repeat all the experiments during the chronic phase of infection (60 days post-infection). The project has already been approved by the Animal Use Ethics Committee of the Federal University of Ouro Preto (CEUA – UFOP). In this phase, we will evaluate biochemical, parasitological, and immune response parameters in mice subjected to diabetes.

**REVIEWER #2**

Abstract

a) In the summary, I suggest changing the information on lines 27 and 28: "However, the impact of a hyperglycemic diet during T. cruzi infection has not been explored in the literature" by "However, the impact of a hyperglycemic diet during T. cruzi infection has been little explored in the literature".

We agree that this sentence could be made clearer, and we have modified it accordingly in the abstract. The revised sentence is now highlighted in green in the manuscript.

b) Originality and importance of the contribution for the development of the field of study. And relevance of:

As mentioned in the article itself, issues involving metabolic disorders and Chagas disease have been little explored in the literature, which makes clear the importance and relevance of this study in the current scenario.

c) Introduction:

- In the introduction, I suggest changing the reference in line 66 (Nunes et al.2018) to WHO (https://dndial.org/doencas/doenca-de-chagas). The current reference cites a study involving Chagas heart disease and would not be the best reference to talk about transmission routes.

Thank you for the suggestion and we have added the correct reference.

"In addition to vector transmission, other routes of infection include congenital transmission, blood transfusion, organ transplantation, oral transmission, and laboratory accidents (DNDi, 2025)."

- The global epidemiological information on Chagas disease presented in lines 68 and 69 must be updated and I suggest using the WHO data available at: (https://www.who.int/news-room/fact-sheets/detail/chagas-disease-(american-trypanosomiasis). Both DNDi and WHO currently report more than 7 million people infected with T. cruzi and more than 100 million people living in risk areas.

"Epidemiologically, CD is considered one of the major neglected tropical diseases, endemic in 21 Latin American countries, with a significant global impact. It is estimated that over 7 million people are affected by the disease, with 100 million at risk of contracting it. However, less than 10% of cases are diagnosed, contributing to an alarming number of disease-related deaths (WHO, 2025)."

d) Methodology

The methodology applied to the study is well written and detailed, which allows for a good understanding of the procedures performed. It is worth highlighting the variety of techniques used. It is necessary to correct the numbering of the topics in line 164: change 2.9.2 to 2.9.1. I also suggest reducing or simplifying the methodology in items 2.11 and 2.12, adding references.

We thank the reviewer for the positive evaluation of our methodology and for the valuable suggestions. We have revised and simplified the descriptions in items 2.11 and 2.12, highlighted in green.

e) Results

- The Figure 1 is not cited in the text.

We appreciate the observation and have added the citation to figure 1 in the text.

"The gating strategy for the phenotypic characterization of total lymphocytes and their subpopulations (CD4+, CD8+, and CD11b+) producing IFN-γ, TNF, and IL-10 is depicted in Figure 1."

- In figure 2A: I suggest changing the colors of the lines by adding very distinct color

Thank you for this observation. We agree with the suggestion to change the colors for better distinction. To ensure consistency and improve the overall clarity of the manuscript, we have decided to update the colors across all figures.

- The Figure 2C is incomplete and it is not possible to verify what is written in the figure.

- The graphs in figure 4 have the abbreviations in the legend in English and on the X axis in Portuguese.

- The graphs in figure 5 have the abbreviations in the legend in English and on the X axis in Portuguese.

We appreciate your observation and have made the correction to the figure, as can be seen in the new figures sent.

- Figures 4A, 4C, 4D and 4F were not cited or described in the text.

We appreciate your observation and have added the citation to the figures in the text, as described below.

"On day 10 post-infection, there were no significant differences in the percentage of IL-10-producing CD4+ (Figure 4-A) or CD8+ T lymphocytes (Figure 4-C), nor in the percentage of IFN-γ-producing CD8+ T lymphocytes (Figure 4-D)."

"However, no significant differences were observed in the percentage of TNF-producing macrophages between the groups (Figure 4-F)."

f) Discussion;

- In line 343 replace Trypanosoma cruzi with T. cruzi.

We appreciate your observation and have made the appropriate correction.

- The discussion presented here is quite superficial and does not correlate the data found in the study. I suggest deepening the discussion, aiming to even extrapolate the observations made in the study with prognostic perspectives in humans.

We recognized that the discussion needed to be revised and rewrote it adding all of the reviewers' suggestions.

g) References;

- The reference Santos et al. 2018 is incomplete.

"Santos, T. A. P. dos. (2018). "Epigenetics in retroperitoneal adipose tissue of rats fed a diet rich in simple carbohydrates after swimming training." Dissertation, Federal University of Ouro Preto. Available from: https://www.repositorio.ufop.br/handle/123456789/10531."

h) Figures and tables.

- The figures and tables are well presented and easy to understand. Some changes/suggestions have been made above.

## SECOND REVIEW ROUND

### REVIEWERS' COMMENTS

### REVIEWER #1

The work has been returned with all the suggested changes, making it clearer and more understandable. The new version has been carefully reviewed, and the key points addressed in the revision have been fully addressed.

### REVIEWER #2

Reviewer did not issue comments.

