## [Reviewer Report · FIRST REVIEW ROUND - REVIEWERS COMMENTS]

## REVIEWER #1

The manuscript is extremely relevant because it demonstrates the impact of a hyperglycemic diet on T. cruzi infection, indicating that it mainly alters parasitological parameters and the regulation of the immune response.

Some modifications should be made to improve the understanding of the manuscript

a) Introduction

1. In the introduction section, line 71, insert a more current reference of epidemiological data.

2. In the introduction section, lines 87-89, “However, studies have demonstrated a link between a hyperglycemic diet and conditions such as obesity and type 2 diabetes mellitus, which are often observed in CD patients”. Insert bibliographic references related to these studies.

3. In the introduction section, lines 100-102, “This would help to better understand the underlying mechanisms of the interaction between diet and parasitic infection, offering insights into potential therapeutic and preventive strategies”. What would these potential therapeutic and preventive strategies be based on studies of the mechanisms of interaction between diet and parasitic infection?

b) Methods

1. Why male C57BL/6 mice were used in this study?

2. In the methods section, lines 113 and 115, “groups: non-infected subjected to a standard diet (NISD) (n=8), infected subjected to a standard diet (ISD) (n=10), non-infected subjected to a hyperglycemic diet (NIHD) (n=8), and infected subjected to a hyperglycemic and infected subjected to a hyperglycemic diet (IHD) (n=10), with half of each group divided for 10 and 30 DPI (days post-infection).” Mention that ISD and NIHD groups were infected by T. cruzi. Was a sample calculation of these groups of animals carried out?

3. Why was the infection performed with the Colombian strain of T. cruzi? The authors believe that the genetic group or DTU of the parasite could interfere with the results of the study?

4. Why was FC receptor blocking not used prior to surface antibody labeling to prevent nonspecific binding?

5. Why did the authors not evaluate the B lymphocyte profile in the spleen?

c) Results

1. In Figure 2B it was not possible to observe the curve referring to the survival rate of the NIHD group.

2. Describe the results obtained in Figures 4A, 4C and 4D even if no statistical differences were observed.

3. In the results section, lines 338 and 339, “Additionally, the percentage of IL-10-producing macrophages was greater in the ISD group compared to its control (NISD) (Figure 5-E).” The sentence is incorrect. The percentage of IL-10-producing macrophages was smaller in the ISD group compared to its control (NISD).

4. In the Figures caption please change the symbol and write the word “bar” as it gives the impression that it is the letter “H”.

5. In the Figure 2C, please change the scale of the graph as it is not possible to verify the groups with low parasite load.

6. Please correct the caption of the Figure 3C. Replace the NISD group with NIHD group.

7. Please correct the caption of the Figure 3F. Replace the NIDH with NIHD and IDH with IHD.

8. In Figure 4 the group acronyms are written incorrectly. Please correct them.

9. In Figure 5 the group acronyms are written incorrectly. Please correct them.

10. In the manuscript text it is written that CD11c was used and in the Table 1 was used CD11b. Would it be CD11b or CD11c?

c) Discussion

1. In the discussion section, lines 343-346. Authors should be careful with these sentences “focusing on its prevalence in Brazil” and “It emphasizes the need to address Chagas disease, a globally neglected condition with significant health impacts”, since the work was carried out on an experimental model and not on humans.

2. In the discussion section, lines 354-358, “While prolonged hyperglycemic diets were expected to induce insulin resistance and hyperglycemia (Jung and Choi, 2017), infection symptoms such as lethargy and reduced appetite likely caused this decrease. Lower food intake and higher parasitemia in the IHD 30 DAI group contributed to the reduced glucose levels.” What is the authors’ hypothesis for the group with IHD having lower food intake that justifies the lower blood glucose levels than those who received the standard diet?

3. In the discussion section, lines 368-370. The authors wrote that “to increased adipose tissue mass, particularly retroperitoneal fat, as observed in non-infected groups (NIHD 10 DAI and NIHD 30 DAI) compared to controls (NISD 10 DIA and NISD 30 DIA)”, but if this sentence refers to the result in Figure 3C, the difference in relation to the NISD group is not demonstrated in the graph.

4. In the discussion section, lines 384-385, “Regarding total serum cholesterol, infected animals on the hyperglycemic diet for 30 DIA (NIHD 30 DAI) […]” Replace the NIHD group with the IHD group.

5. In the discussion section, line 386, “[…] levels compared to non-infected animals on the same diet for the same period (IHD 30 DAI). Replace the IHD group with the NIHD group.

6. In the discussion section, lines 388 to 390, “This may explain the lower cholesterol levels in the NIHD 30 DAI group compared to the IHD 30 DAI group, as the NIHD 30 DAI group had a lower weekly intake of the diet.” The sentence is incorrect. Figure 3F demonstrates higher cholesterol levels in the NIHD 30 DAI group compared to the IHD 30 DAI group. Furthermore, Figure 3A demonstrated that the NIHD 30 DAI group had a higher food intake than the IHD 30 DAI group.

7. Days 86 and 66 refer to 10 DAI and 30 DAI, respectively? This data is not shown in the figures.

8. In the discussion section, lines 412, “Parasitic loads were higher in the heart on day 86 compared to day 66 in both infected groups, likely due to the heart tropism of the Colombian strain.”. If this sentence is referring to Figure 3C, it is incorrect, as the IHD group presented a higher parasite load on the 10 DAI and the ISD group on the 30 DAI in the heart.

9. In the discussion section, lines 419 and 420, “[…] with no differences in other groups or liver tissue.” No analysis was performed on the liver. The sentence is incorrect.

10. What are the perspectives of the study? The authors intend to perform biochemical, parasitological and immune response analyses in mice in the chronic phase, that is, with a longer time of T. cruzi infection?

## REVIEWER #2

Influence of a Hyperglycemic Diet on Trypanosoma cruzi Infection in Mice Model

a) Adequacy of the abstract;

The summary is well structured and provides basic and fundamental information about the content that will be covered in the article.

- In the summary, I suggest changing the information on lines 27 and 28: “However, the impact of a hyperglycemic diet during T. cruzi infection has not been explored in the literature” by “However, the impact of a hyperglycemic diet during T. cruzi infection has been little explored in the literature”.

b) Originality and importance of the contribution for the development of the field of study. And relevance of:

As mentioned in the article itself, issues involving metabolic disorders and Chagas disease have been little explored in the literature, which makes clear the importance and relevance of this study in the current scenario.

c) Introduction:

- In the introduction, I suggest changing the reference in line 66 (Nunes et al.2018) to WHO (https://dndial.org/doencas/doenca-de-chagas).. The current reference cites a study involving Chagas heart disease and would not be the best reference to talk about transmission routes.

- The global epidemiological information on Chagas disease presented in lines 68 and 69 must be updated and I suggest using the WHO data available at: (https://www.who.int/news-room/fact-sheets/detail/chagas-disease-(american-trypanosomiasis). Both DNDi and WHO currently report more than 7 million people infected with T. cruzi and more than 100 million people living in risk areas.

d) Methodology

The methodology applied to the study is well written and detailed, which allows for a good understanding of the procedures performed. It is worth highlighting the variety of techniques used. It is necessary to correct the numbering of the topics in line 164: change 2.9.2 to 2.9.1. I also suggest reducing or simplifying the methodology in items 2.11 and 2.12, adding references.

e) Results

- The Figure 1 is not cited in the text.

- In figure 2A: I suggest changing the colors of the lines by adding very distinct color.

- The Figure 2C is incomplete and it is not possible to verify what is written in the figure.

- The graphs in figure 4 have the abbreviations in the legend in English and on the X axis in Portuguese.

- Figures 4A, 4C, 4D and 4F were not cited or described in the text.

- The graphs in figure 5 have the abbreviations in the legend in English and on the X axis in Portuguese.

f) Discussion

- In line 343 replace Trypanosoma cruzi with T. cruzi.

- The discussion presented here is quite superficial and does not correlate the data found in the study. I suggest deepening the discussion, aiming to even extrapolate the observations made in the study with prognostic perspectives in humans.

g) References

- The reference Santos et al. 2018 is incomplete.

h) Figures and tables.

- The figures and tables are well presented and easy to understand. Some changes/suggestions have been made above.

---

## [Author Response · AUTHORS RESPONSE TO REVIEWERS]

## To: Dr. Adeilton Brandão, Ph. D.

Handling Editor

Memórias do Instituto Oswaldo Cruz

Ref: MIOC-2025-0092

Ouro Preto, June, 2025

Dear Dr. Adeilton Brandão,

We have received the e-mail containing the comments from the Editor and Reviewers regarding our manuscript (MIOC-2025-0092), entitled: “Influence of a Hyperglycemic Diet on Trypanosoma cruzi Infection in a Mouse Model” by Merces et al., submitted to Memórias do Instituto Oswaldo Cruz.

We carefully addressed all the points raised by the Editor and Reviewers, responding to each comment in detail to improve the quality of our manuscript and ensure it meets the standards for publication in Memórias do Instituto Oswaldo Cruz.

Below, we present the responses to all the queries raised by the Editor and Reviewers. The corresponding modifications in the manuscript are highlighted in green. As requested, our replies to each reviewer’s comment are provided below, formatted in “Arial Black” font.

---

## [Reviewer Report · REVIEWERS COMMENTS]

## We thank Reviewer #1 and Reviewer #2 for their careful review of our manuscript. Our responses to their questions are provided below: REVIEWER #1

Introduction

1. In the introduction section, line 71, insert a more current reference of epidemiological data.

Thank you for the suggestion and we have added a more up-to-date reference.

“It is estimated that over 7 million people are affected by the disease, with 100 million at risk of contracting it. However, less than 10% of cases are diagnosed, contributing to an alarming number of disease-related deaths (WHO, 2025).”

2. In the introduction section, lines 87-89, “However, studies have demonstrated a link between a hyperglycemic diet and conditions such as obesity and type 2 diabetes mellitus, which are often observed in CD patients”. Insert bibliographic references related to these studies.

“However, studies have demonstrated a link between a hyperglycemic diet and conditions such as obesity and type 2 diabetes mellitus, which are often observed in CD patients. For example, research assessing the prevalence of metabolic syndrome in CD patients revealed high rates of obesity, diabetes mellitus, and dyslipidemia in this population (Xavier et al., 2019).”

3. In the introduction section, lines 100-102, “This would help to better understand the underlying mechanisms of the interaction between diet and parasitic infection, offering insights into potential therapeutic and preventive strategies”. What would these potential therapeutic and preventive strategies be based on studies of the mechanisms of interaction between diet and parasitic infection?

Since a hyperglycemic diet is associated with an inflammatory state that could exacerbate tissue damage caused by parasitic infection, the implementation of specific dietary guidelines could help attenuate disease progression in infected individuals. Given that adipose tissue serves as a reservoir for T. cruzi and plays a key role in modulating the immune response, pharmacological agents that improve metabolic control, such as insulin sensitizers and lipid-lowering drugs, could be employed as adjuncts to etiological treatment with benznidazole, particularly in Chagas disease patients with metabolic syndrome.

Furthermore, in endemic areas with a higher prevalence of individuals affected by diabetes, hypertension, and obesity, a better understanding of the correlation between metabolic syndrome and the course of T. cruzi infection may be essential for developing personalized prevention programs that integrate both disease control and improved management of metabolic syndrome.

Methods

1. Why male C57BL/6 mice were used in this study?

We chose male C57BL/6 mice because they are an isogenic strain available at our Animal Science Center, allowing us to minimize genetic variability in the study. Additionally, this strain exhibits partial resistance to Trypanosoma cruzi infection, which helps ensure that the inflammatory response generated by the infection itself does not overshadow the effects of the diet.

2. In the methods section, lines 113 and 115, “groups: non-infected subjected to a standard diet (NISD) (n=8), infected subjected to a standard diet (ISD) (n=10), non-infected subjected to a hyperglycemic diet (NIHD) (n=8), and infected subjected to a hyperglycemic and infected subjected to a hyperglycemic diet (IHD) (n=10), with half of each group divided for 10 and 30 DPI (days post-infection).” Mention that ISD and NIHD groups were infected by T. cruzi. Was a sample calculation of these groups of animals carried out?

Only the ISD and IHD groups were infected with Trypanosoma cruzi. The NIHD group was not infected; it was only subjected to the hyperglycemic diet. We will clarify this point in the revised version of the manuscript to avoid any misunderstanding.

“The animals were randomly assigned to four experimental groups: non-infected animals subjected to a standard diet (NISD, n = 8), T. cruzi-infected animals subjected to a standard diet (ISD, n = 10), non-infected animals subjected to a hyperglycemic diet (NIHD, n = 8), and T. cruzi-infected animals subjected to a hyperglycemic diet (IHD, n = 10). Each group was further subdivided equally for evaluation at 10- and 30-days post-infection (DPI).”

The sample size was calculated using BioEstat software, version 5.3, which is widely used for statistical analyses in biological studies. The calculation considered a minimum detectable difference of 50 units between treatment means, with a standard error deviation of 20. The analysis included four treatment groups, with a statistical power of 95% and a significance level (α) of 0.05. Additionally, potential mortality in the infected groups due to the infection, as well as natural mortality in the non-infected groups, was taken into account during the sample size estimation.

3. Why was the infection performed with the Colombian strain of T. cruzi? The authors believe that the genetic group or DTU of the parasite could interfere with the results of the study?

The Colombian strain was chosen due to its ability to induce a chronic infection in mice, which allows for the evaluation of long-term effects of a hyperglycemic diet. This strain is known to promote a clinical condition that mimics Chagas disease in murine models, including sustained inflammation, fibrosis and oxidative stress in the heart. And as we intended to evaluate the impact of the hyperglycemic diet on the course of the infection, we chose this strain due to the better survival of the animals when infected with an intermediate inoculum by this strain.

4. Why was FC receptor blocking not used prior to surface antibody labeling to prevent nonspecific binding?

In all the flow cytometry experiments carried out in this study, to avoid non-specific labeling, we diluted the antibodies in a solution containing normal serum (3-5%) from the host in which the antibody was produced. In this way, we ensured that the markings observed in the experiments were linked to the identification of the cell populations evaluated and that there were no non-specific markings. In this way, we were able to guarantee the quality of the experiments and the markings, as well as reducing the costs involved in immunophenotyping experiments.

5. Why did the authors not evaluate the B lymphocyte profile in the spleen?

In this first study, we chose to evaluate the cellular immune response, since hyperglycemia mainly affects this kind of immune response. These mechanisms include suppression of cytokine production, defects in phagocytosis, dysfunction of T cells, and leukocyte recruitment inhibition. However, in future studies we intend to expand the assessments to include other cells involved in the humoral immune response, although it is less affected by the hyperglycemic state.

Results

1. In Figure 2B it was not possible to observe the curve referring to the survival rate of the NIHD group.

We appreciate the critical analysis and have made the appropriate modifications to better visualize the data, as demonstrated in the new submitted figure 2.

2. Describe the results obtained in Figures 4A, 4C and 4D even if no statistical differences were observed.

We appreciate the critical analysis and have included the following description in the text:

“On day 10 post-infection, there were no significant differences in the percentage of IL-10-producing CD4⁺ (Figure 4-A) or CD8⁺ T lymphocytes (Figure 4-C), nor in the percentage of IFN-γ-producing CD8⁺ T lymphocytes (Figure 4-D).”

3. In the results section, lines 338 and 339, “Additionally, the percentage of IL-10-producing macrophages was greater in the ISD group compared to its control (NISD) (Figure 5-E).” The sentence is incorrect. The percentage of IL-10-producing macrophages was smaller in the ISD group compared to its control (NISD).

We appreciate your critical review and have made appropriate modifications.

“Additionally, the percentage of IL-10-producing macrophages was smaller in the ISD group when compared to its control (NISD) (Figure 5-E).”

4. In the Figures caption please change the symbol and write the word “bar” as it gives the impression that it is the letter “H”.

We appreciate your critical review and have made appropriate modifications in the figures caption.

5. In the Figure 2C, please change the scale of the graph as it is not possible to verify the groups with low parasite load.

We appreciate your critical review and have made appropriate modifications in the scale of the graph.

6. Please correct the caption of the Figure 3C. Replace the NISD group with NIHD group.

7. Please correct the caption of the Figure 3F. Replace the NIDH group with NIHD group and IDH with IHD.

8. In Figure 4 the group acronyms are written incorrectly. Please correct them.

9. In Figure 5 the group acronyms are written incorrectly. Please correct them.

We appreciate your critical review and have made appropriate modifications.

10. In the manuscript text it is written that CD11c was used and in the Table 1 was used CD11b. Would it be CD11b or CD11c?

We are thankful for this observation. The table contains the correct information: the marker used was CD11b. The correction has been made in the manuscript and is highlighted in green.

Discussion

1. In the discussion section, lines 343-346. Authors should be careful with these sentences “focusing on its prevalence in Brazil” and “It emphasizes the need to address Chagas disease, a globally neglected condition with significant health impacts”, since the work was carried out on an experimental model and not on humans.

We agree that this sentence could lead to misunderstandings, as the study was conducted in an experimental model. Therefore, we decided to revise the entire sentence to ensure greater clarity.

2. In the discussion section, lines 354-358, “While prolonged hyperglycemic diets were expected to induce insulin resistance and hyperglycemia (Jung and Choi, 2017), infection symptoms such as lethargy and reduced appetite likely caused this decrease. Lower food intake and higher parasitemia in the IHD 30 DAI group contributed to the reduced glucose levels.” What is the authors’ hypothesis for the group with IHD having lower food intake that justifies the lower blood glucose levels than those who received the standard diet?

As the IHD group had higher parasitemia around 30 DAI compared to the ISD group, this may have been associated with lethargy, fever due to worsening of the parasitological condition and consequently reduced food intake, resulting in lower blood glucose levels.

3. In the discussion section, lines 368-370. The authors wrote that “to increased adipose tissue mass, particularly retroperitoneal fat, as observed in non-infected groups (NIHD 10 DAI and NIHD 30 DAI) compared to controls (NISD 10 DIA and NISD 30 DIA)”, but if this sentence refers to the result in Figure 3C, the difference in relation to the NISD group is not demonstrated in the graph.

We appreciate this observation. The difference in retroperitoneal fat was observed within the NIHD group when comparing the two time points (10 and 30 DAI), and not between NIHD and NISD groups, as previously stated. We have corrected the sentence in the manuscript to accurately reflect this result shown in Figure 3C.

4. In the discussion section, lines 384-385, “Regarding total serum cholesterol, infected animals on the hyperglycemic diet for 30 DIA (NIHD 30 DAI) […]” Replace the NIHD group with the IHD group.

We appreciate the critical analysis and we have already made the correction, as highlighted in the text.

“Regarding total serum cholesterol, infected animals on the hyperglycemic diet at 30 DAI (IHD 30 DAI) had lower levels compared to non-infected animals on the same diet in the same period (NIHD 30 DAI).”

5. In the discussion section, line 386, “[…] levels compared to non-infected animals on the same diet for the same period (IHD 30 DAI). Replace the IHD group with the NIHD group.

We appreciate the critical analysis and we have already made the correction, as highlighted in the text.

“Regarding total serum cholesterol, infected animals on the hyperglycemic diet at 30 DAI (IHD 30 DAI) had lower levels compared to non-infected animals on the same diet in the same period (NIHD 30 DAI).”

6. In the discussion section, lines 388 to 390, “This may explain the lower cholesterol levels in the NIHD 30 DAI group compared to the IHD 30 DAI group, as the NIHD 30 DAI group had a lower weekly intake of the diet.” The sentence is incorrect. Figure 3F demonstrates higher cholesterol levels in the NIHD 30 DAI group compared to the IHD 30 DAI group. Furthermore, Figure 3A demonstrated that the NIHD 30 DAI group had a higher food intake than the IHD 30 DAI group.

Thank you for your observation. The description of the results was inaccurate. We have revised the text to correctly reflect that, at 30 DAI, the NIHD group had higher cholesterol levels and higher food intake compared to the IHD group. We modified the description to emphasize that, although hyperglycemic diets are typically associated with increased cholesterol levels, in the presence of infection, reduced food intake likely contributed to the lower cholesterol levels observed in the IHD group. The manuscript has been updated accordingly.

7. Days 86 and 66 refer to 10 DAI and 30 DAI, respectively? This data is not shown in the figures.

Yes, days 86 and 66 correspond to 10 and 30 days after infection (DAI), respectively. The text has been revised to clarify this information and prevent any misunderstanding.

8. In the discussion section, lines 412, “Parasitic loads were higher in the heart on day 86 compared to day 66 in both infected groups, likely due to the heart tropism of the Colombian strain.”. If this sentence is referring to Figure 3C, it is incorrect, as the IHD group presented a higher parasite load on the 10 DAI and the ISD group on the 30 DAI in the heart.

Thank you for your comment. As indicated in item 5 of the results, we have modified the scale of Figure 2C to improve comprehension. With this adjustment, it is clear that parasitic loads in the heart increased from 10 to 30 DAI in both infected groups, supporting the correctness of the sentence in the manuscript.

9. In the discussion section, lines 419 and 420, “[…] with no differences in other groups or liver tissue.” No analysis was performed on the liver. The sentence is incorrect.

We have removed this sentence from the manuscript. Although the liver was collected and the analysis is currently ongoing, the results are not presented in this study.

10. What are the perspectives of the study? The authors intend to perform biochemical, parasitological and immune response analyses in mice in the chronic phase, that is, with a longer time of T. cruzi infection?

Yes, we intend to repeat all the experiments during the chronic phase of infection (60 days post-infection). The project has already been approved by the Animal Use Ethics Committee of the Federal University of Ouro Preto (CEUA – UFOP). In this phase, we will evaluate biochemical, parasitological, and immune response parameters in mice subjected to diabetes.

Reviewer #2

Abstract

a) In the summary, I suggest changing the information on lines 27 and 28: “However, the impact of a hyperglycemic diet during T. cruzi infection has not been explored in the literature” by “However, the impact of a hyperglycemic diet during T. cruzi infection has been little explored in the literature”.

We agree that this sentence could be made clearer, and we have modified it accordingly in the abstract. The revised sentence is now highlighted in green in the manuscript.

b) Originality and importance of the contribution for the development of the field of study. And relevance of:

As mentioned in the article itself, issues involving metabolic disorders and Chagas disease have been little explored in the literature, which makes clear the importance and relevance of this study in the current scenario.

c) Introduction:

- In the introduction, I suggest changing the reference in line 66 (Nunes et al.2018) to WHO (https://dndial.org/doencas/doenca-de-chagas). The current reference cites a study involving Chagas heart disease and would not be the best reference to talk about transmission routes.

Thank you for the suggestion and we have added the correct reference.

“In addition to vector transmission, other routes of infection include congenital transmission, blood transfusion, organ transplantation, oral transmission, and laboratory accidents (DNDi, 2025).”

- The global epidemiological information on Chagas disease presented in lines 68 and 69 must be updated and I suggest using the WHO data available at: (https://www.who.int/news-room/fact-sheets/detail/chagas-disease-(american-trypanosomiasis). Both DNDi and WHO currently report more than 7 million people infected with T. cruzi and more than 100 million people living in risk areas.

“Epidemiologically, CD is considered one of the major neglected tropical diseases, endemic in 21 Latin American countries, with a significant global impact. It is estimated that over 7 million people are affected by the disease, with 100 million at risk of contracting it. However, less than 10% of cases are diagnosed, contributing to an alarming number of disease-related deaths (WHO, 2025).”

d) Methodology

The methodology applied to the study is well written and detailed, which allows for a good understanding of the procedures performed. It is worth highlighting the variety of techniques used. It is necessary to correct the numbering of the topics in line 164: change 2.9.2 to 2.9.1. I also suggest reducing or simplifying the methodology in items 2.11 and 2.12, adding references.

We thank the reviewer for the positive evaluation of our methodology and for the valuable suggestions. We have revised and simplified the descriptions in items 2.11 and 2.12, highlighted in green.

e) Results

- The Figure 1 is not cited in the text.

We appreciate the observation and have added the citation to figure 1 in the text.

“The gating strategy for the phenotypic characterization of total lymphocytes and their subpopulations (CD4⁺, CD8⁺, and CD11b⁺) producing IFN-γ, TNF, and IL-10 is depicted in Figure 1.”

- In figure 2A: I suggest changing the colors of the lines by adding very distinct color

Thank you for this observation. We agree with the suggestion to change the colors for better distinction. To ensure consistency and improve the overall clarity of the manuscript, we have decided to update the colors across all figures.

- The Figure 2C is incomplete and it is not possible to verify what is written in the figure.

- The graphs in figure 4 have the abbreviations in the legend in English and on the X axis in Portuguese.

- The graphs in figure 5 have the abbreviations in the legend in English and on the X axis in Portuguese.

We appreciate your observation and have made the correction to the figure, as can be seen in the new figures sent.

- Figures 4A, 4C, 4D and 4F were not cited or described in the text.

We appreciate your observation and have added the citation to the figures in the text, as described below.

“On day 10 post-infection, there were no significant differences in the percentage of IL-10-producing CD4⁺ (Figure 4-A) or CD8⁺ T lymphocytes (Figure 4-C), nor in the percentage of IFN-γ-producing CD8⁺ T lymphocytes (Figure 4-D).”

“However, no significant differences were observed in the percentage of TNF-producing macrophages between the groups (Figure 4-F).”

f) Discussion;

- In line 343 replace Trypanosoma cruzi with T. cruzi.

We appreciate your observation and have made the appropriate correction.

- The discussion presented here is quite superficial and does not correlate the data found in the study. I suggest deepening the discussion, aiming to even extrapolate the observations made in the study with prognostic perspectives in humans.

We recognized that the discussion needed to be revised and rewrote it adding all of the reviewers’ suggestions.

g) References;

- The reference Santos et al. 2018 is incomplete.

“Santos, T. A. P. dos. (2018). “Epigenetics in retroperitoneal adipose tissue of rats fed a diet rich in simple carbohydrates after swimming training.” Dissertation, Federal University of Ouro Preto. Available from: https://www.repositorio.ufop.br/handle/123456789/10531.”

h) Figures and tables.

- The figures and tables are well presented and easy to understand. Some changes/suggestions have been made above.

---

## [Reviewer Report · REVIEWERS COMMENTS]

## REVIEWER #1

The work has been returned with all the suggested changes, making it clearer and more understandable. The new version has been carefully reviewed, and the key points addressed in the revision have been fully addressed.

## REVIEWER #2

Reviewer did not issue comments.